# In vitro hepatic metabolism of mefloquine using microsomes from cats, dogs and the common brush-tailed possum (*Trichosurus vulpecula*)

**Aaron Michael Izes**[ID][◔], **Benjamin Kimble**[◔], **Jacqueline Marie Norris**[◔], **Merran Govendir**[ID]*[◔]

Sydney School of Veterinary Science, Faculty of Science, The University of Sydney, Sydney, NSW, Australia

◔ These authors contributed equally to this work.
* merran.govendir@sydney.edu.au

**Data Availability Statement:** All relevant data are within the paper and its Supporting Information files.

## Abstract

Feline infectious peritonitis (FIP) is a systemic, fatal, viral-induced, immune-mediated disease of cats caused by feline infectious peritonitis virus (FIPV). Mefloquine, a human anti-malarial agent, has been shown to inhibit FIPV in vitro. As a first step to evaluate its efficacy and safety profile as a potential FIP treatment for cats, mefloquine underwent incubation in feline, canine and common brush-tailed possum microsomes and phase I metabolism cofactors to determine its rate of phase I depletion. Tramadol was used as a phase I positive control as it undergoes this reaction in both dogs and cats. Using the substrate depletion method, the in vitro intrinsic clearance (mean ± S.D.) of mefloquine by pooled feline and common brush-tailed possum microsomes was 4.5 ± 0.35 and 18.25 ± 3.18 µL/min/mg protein, respectively. However, phase I intrinsic clearance was too slow to determine with canine microsomes. Liquid chromatography—mass spectrometry (LC-MS) identified carboxymefloquine in samples generated by feline microsomes as well as negative controls, suggesting some mefloquine instability. Mefloquine also underwent incubation with feline, canine and common brush-tailed possum microsomes and phase II glucuronidative metabolism cofactors. O-desmethyltramadol (ODMT or M1) was used as a positive control as it undergoes a phase II glucuronidation reaction in these species. The rates of phase II mefloquine depletion by microsomes by all three species were too slow to estimate. Therefore mefloquine likely undergoes phase I hepatic metabolism catalysed by feline and common brush-tailed possum microsomes but not phase II glucuronidative metabolism in all three species and mefloquine is not likely to have delayed elimination in cats with clinically normal, hepatic function.

## Introduction

Feline infectious peritonitis is a systemic, fatal, viral-induced immune-mediated disease primarily affecting younger cats [1, 2]. It is caused by a virulent biotype of feline coronaviruses

**Funding:** M.G. and J.M.N received the following awards from the following funders to undertake this research: Winn Feline Foundation 16-023 Australian Companion Animal Health Foundation 005/2016 Feline Health Research Foundation University of Sydney IRMA number 183456.

**Competing interests:** The authors have declared that no competing interests exist.

(FCoV) known as FIPV. Previously used therapies for FIP have either relied on immunosuppressive agents to dampen clinical signs temporarily, or attempted to repurpose human antiviral treatments, with neither approach achieving lasting success [3]. Recently, therapeutic drugs have been identified that address the underlying problem of feline coronavirus replication [4, 5]. However, veterinary access to these antiviral treatments is currently problematic. Consequently, additional inexpensive, safe, antiviral treatments, such as those accessible by prescription, require investigation for their potential to treat FIPV infected cats. The demand for efficacious antivirals against FIPV is crucial both for reducing clinical disease with minimal adverse effects, as well as the potential for preventing individual treatments selecting for viral resistance.

The antimalarial drug mefloquine is currently used for both prevention (as a monotherapy) and treatment (either alone, or in combination with artesunate) of chloroquine-resistant *Plasmodium falciparum* malaria in humans [6–8]. It was found to substantially reduce the viral load of FIPV and feline calicivirus (FCV) at low concentrations in infected Crandell Rees feline kidney cells without cytotoxic effects [9, 10]. This observation suggested that mefloquine may be useful when administered to cats with FIP or FCV related disease [9]. However, mefloquine undergoes substantial hepatic metabolism in people [11, 12]. As cats lack several major phenol UDP-glucuronosyltransferase (UGT) enzymes required for phase II glucuronidative metabolism [13, 14], it is prudent to obtain information about mefloquine's intrinsic clearance in the cat before conducting clinical trials in the live animal. Although there are many models by which to assess in vitro metabolism, microsomes contain many (but not all) hepatic metabolic enzymes. Therefore, the aim of this study was to ascertain mefloquine's in vitro intrinsic clearance (in vitro Cl$_{int}$) when incubated with feline hepatic microsomes with respect to both phase I (e.g., oxidation, hydrolysis or reduction reactions) and phase II (e.g., conjugative glucuronidation) metabolism. Likewise, mefloquine was also incubated with canine microsomes for both phase I and phase II metabolism reactions in order to compare the rate of in vitro clearance of mefloquine between feline and canine microsomes. Mefloquine's phase I metabolism by canine microsomes has been previously reported [11] and in contrast to cats, dogs are not known to experience delayed phase II conjugative metabolism. Mefloquine was also incubated with microsomes from the common brush-tailed possum (*Trichosurus vulpecula*) as this species of possum is recognised as a rapid xenobiotic metaboliser [15].

## Materials and methods

### Chemicals and materials

Mefloquine, verapamil (as the internal standard [IS] for high pressure liquid chromatography [HPLC]), tramadol hydrochloride, ODMT, phase I enzyme co-factor β-nicotinamide adenine dinucleotide phosphate sodium (NADP), glucose 6-phosphate, catalytic enzyme glucose 6-phosphate dehydrogenase, phase II enzyme cofactor uridine 5'-diphosphoglucuronic acid disodium salt (UDPGA) and alamethicin (a pore-forming peptide antibiotic used to enhance cofactor availability [14]) were purchased from Sigma-Aldrich (Castle Hill, NSW, Australia). HPLC grade acetonitrile, HPLC grade methanol and ethyl acetate were purchased from Thermo Fisher Scientific (Macquarie Park, NSW, Australia). Hepatic microsomes from pooled male cats and pooled male beagles were purchased from BioreclamationIVT (Frankfurt, Germany; product numbers S00709 and M00201, respectively), shipped on liquid nitrogen and stored at -80˚C prior to use. The common brush-tailed possum microsomes (2 females and one male) were extracted and prepared according to Hill (2003) [16] and modified by Kimble *et al*. (2014) [15]. Prior to liver collection, possums were euthanized with pentobarbitone for humane reasons independent of this study. These animals demonstrated no outward clinical

signs of hepatic disease and their livers appeared grossly unremarkable on post-mortem examination. Liver collection was approved by The University of Sydney Animal Ethics Committee (AEC protocol: 2013/6062). Commercially procured feline and canine microsomes had a hepatic microsomal protein concentration of 20 mg/mL as did those of the common brush-tailed possum microsomes. The activity of the canine and feline microsomes as determined by the manufacturer are presented in the Supplementary information file (S1 Table). The common brush-tailed possum hepatic microsomal protein concentration was measured using a Bradford assay kit (Bio-Rad, Hercules, CA, USA) standardized with bovine serum albumin [17]. The pooled hepatic feline and canine microsomes contained enzymes from three individuals.

### Hepatic microsomal incubations

For both phase I and phase II reactions, the substrate depletion method was used to estimate the in vitro $Cl_{int}$ [18]. The respective microsomal protein concentrations were chosen after undertaking preliminary assay condition optimisation trials [19] based on recommended best practices for microsomal experimentation [20, 21]. Incubations times were selected as those within which the rate of depletion was linear and therefore presumed to follow first order elimination [21].

### Phase I microsomal incubation

The phase I hepatic microsomal assay was modified from a method described by Kimble *et al.* (2014) [15]. Mefloquine (10 μM) was pre-incubated in 4.0 mL of 0.1 M sodium phosphate buffer (pH 7.4) containing a NADPH regenerating system (1.0 mM NADP, 0.8 U glucose 6-phosphate dehydrogenase and 3.0 mM glucose 6-phosphate) and 3.0 mM magnesium chloride ($MgCl_2$), in an open air shaking water bath at 37˚C for five minutes. The 10 μM final concentration of mefloquine was selected as it was the same concentration used by McDonagh *et al.* (2014) in FIPV infected Crandell Rees feline kidney cells [10]. The enzymatic reaction was initiated by adding a 1.0 mg/mL concentration of either feline or canine microsomal protein and 0.5 mg/mL of brush-tailed possum microsomal protein. Collectively, the microsomes, enzyme co-factors and solvents are referred to as a "matrix". During incubation the matrix was gently agitated with the pipette tip and 500 μL aliquots were removed at time (t) = 0, 30, 60 and 90 minutes. For the possum incubation aliquots were removed at t = 0, 30 and 60 minutes. Each extracted aliquot was mixed with 250 μL of ice-cold methanol (which also contained 10 μM of the IS) to deactivate the reaction. Once the reaction was deactivated, mefloquine samples underwent liquid-liquid extraction whereby 200 μL of each sample was added to 1.6 mL of ethyl acetate, vortexed and then centrifuged at 14,000 × g for five minutes. The organic layer was then transferred to a new 2.0 mL microcentrifuge tube and dried in a Thermo Fisher Scientific Speed Vac. concentrator (Macquarie Park, NSW, Australia) at 30˚C for one hour. The dried residue was reconstituted in 100 μL of 50% methanol. The reconstituted samples were vortexed and centrifuged at 14,000 × g for five minutes to remove any particulates. The supernatant was injected into the HPLC system. All samples were run in duplicate.

Negative controls underwent the incubation process described above and included one set that contained all incubation components except the substrate, another set which omitted the NADPH cofactors and another which omitted the microsomes. The positive control substrate was tramadol (1 μM tramadol incubated with 1.0 mg/mL of either feline microsomes or canine microsomes over 60 minutes) as it undergoes phase I metabolism to ODMT (M1) and to N-desmethyltramadol (M2 or NDMT) when incubated with canine microsomes and to ODMT only when incubated with feline microsomes [22]. The tramadol samples did not undergo liquid-liquid extraction and no internal standard was used.

## Phase II microsomal incubation

The phase II hepatic microsomal assay was modified from a method described by Slovak *et al.* (2017) [14]. Mefloquine (5.0 μM final concentration) was pre-incubated in 0.5 mL of 50 mM potassium phosphate buffer (pH 7.4) containing 2.5 μg/μL alamethicin and 0.5 mg/mL concentrations of hepatic microsomal protein from either cats, dogs or common brush-tailed possums, in an open-air shaking water bath at 37˚C for five minutes. The enzymatic reaction was initiated by adding a solution consisting of 100 mM potassium phosphate buffer, purified water, 5.0 mM UDPGA and 0.64 mL of 50 mM $MgCl_2$. During the incubation the matrix was gently agitated with the pipette tip and 100 μL aliquots were removed at time (t) = 0, 60, 180 and 360 minutes. Each extracted aliquot was mixed with 100 μL of ice-cold methanol (which also contained 10 μM of IS) to deactivate the reaction. Once the reaction was deactivated, each sample with mefloquine as the substrate underwent liquid-liquid extraction as per the phase I reaction. The organic layer was then transferred to a new 2.0 mL microcentrifuge tube and dried in the concentrator at 30˚C for one hour. The dried residue was reconstituted in 100 μL of 50% methanol. The reconstituted samples were vortexed and centrifuged at $14,000 \times g$ for five minutes to remove any particulates. The supernatant was injected into the HPLC system. All samples, including controls, were run as duplicates.

A negative control group for each of the three species utilised the same incubation protocol as described above. However, UDPGA was not added to the matrix. The positive control involved the depletion of ODMT (the tramadol metabolite known as M1) to ODMT glucuronide, the product of a recognised phase II glucuronidation reaction [23], generated when incubated with canine and common brush-tailed possum microsomes and minimally with feline microsomes [19]. The chromatographic conditions to detect ODMT depletion to ODMT glucuronide were the same as those that detected tramadol depletion in the phase I incubation. The ODMT samples did not undergo liquid-liquid extraction and no internal standard was used.

## HPLC chromatographic conditions for the phase I and phase II reaction

Mefloquine and verapamil were identified in the microsomal matrix through an HPLC method modified from that of Gbotosho *et al.* (2012) [24]. The HPLC system consisted of a Shimadzu LC-20AT delivery unit, DGU-20A3 HT degassing solvent delivery unit, SIL-20A auto injector, SPD-20A UV detector and CTO-20A column oven. Shimadzu LC Solution software (Kyoto, Japan) was used for chromatographic control, data collection and data processing. Chromatographic separation was performed with a Microsorb-MV C18 column (250 mm × 4.6 mm i.d., 5.0 μm; Varian, Mulgrave, VIC, Australia) with a 1.0 mm Optic-guard C18 pre-column (Choice Analytical, Thornleigh, NSW, Australia) under a pressure of 1,900 psi at 25˚C. The isocratic mobile phase consisted of a mixture of 25 mM sodium phosphate buffer with 1.0% triethylamine adjusted to pH 5.5: acetonitrile: methanol (35:35:30, v/v/v) at a flow rate of 1.0 mL/min. The injection volume was 10 μL per sample. The diode array detector was set at a wavelength of 220 nm. The total run time for each sample was 21 minutes. The retention times of verapamil and mefloquine were 3.6 and 4.3 minutes, respectively.

Detection of tramadol depletion for the phase I reaction and depletion of ODMT to ODMT glucuronide to demonstrate the phase II reaction used the same HPLC system as described previously but chromatographic separation involved a Luna C18 column (250 mm x 4.6 mm i. d., 5.0 μm; Phenomenex, Lane Cove, NSW, Australia) with a 1.0 mm Optic-guard C18 pre-column (Choice Analytical, Thornleigh, NSW, Australia) under a pressure of 1,800 psi at ambient room temperature (25˚C). The isocratic mobile phase consisted of a mixture of 0.01M phosphate buffer and acetonitrile (82.5:17.5, v/v), containing 0.1% triethylamine and then adjusted

to pH 5.5, at a flow rate of 1.0 mL/min. The injection volume was 10 μL. Fluorescence detection (excitation: 200 nm / emission: 301 nm) was used. The total run time for each sample was 22.5 minutes and the retention times of ODMT and tramadol were approximately 6.7 and 17.2 minutes, respectively [22]. Each incubation was run in duplicate during a single experiment.

## Statistical analyses

Statistical data analysis was performed using GraphPad Prism 8.1.1 software (Graph Pad Software, Inc., CA, USA). The substrate depletion model is not reliable when the depletion is < 15% during the incubation period [20]. However if the depletion was greater than 15% then a two way ANOVA was performed to compare the mean depletion (%) between species at each time-point. If the ANOVA was significant Tukey's multiple comparison tests were undertaken. Statistical significance was accepted at $P < 0.05$.

## LC-MS chromatographic conditions

An unknown HPLC chromatogram peak was evident in all phase I mefloquine samples when incubated with feline microsomes. Consequently, samples were re-run and captured at the retention time when this peak was visualised on chromatograms. This unknown peak was hypothesised to be carboxymefloquine. Therefore, a modified LC-MS method [25, 26]from Geditz et al., 2014 [25] utilised a pseudo-targeted approach to try to identify carboxymefloquine by its mass-to-charge ratio (*m/z*) value of 308.01. The LC-MS equipment consisted of a Thermo Scientific Vanquish UHPLC system (Thermo Fisher Scientific, San Jose, CA, USA) connected to a hybrid quadrupole orbitrap mass spectrometer equipped with an electrospray ionization (ESI) source (Thermo Scientific Q Exactive HF-X Hybrid Quadrupole-Orbitrap mass spectrometer; Thermo Fisher Scientific, San Jose, CA, USA). An Agilent Zorbax Extend C18 column (50 mm × 2.1 mm i.d., 1.8 μm; Agilent Technologies Australia, Mulgrave, VIC, Australia) was used for chromatographic separation and Thermo X-calibur software version 4.1 (Thermo Fisher Scientific, San Jose, CA, USA) was used for instrument control and data processing. The mass spectrometer was operated in negative ion mode. The isocratic mobile phase consisted of methanol and water (+ 0.1% formic acid) (50:50 v/v). The run time was 10 minutes with a flow rate of 0.35 mL/min. The injection volume was 1.0 μL and the column temperature was set at 45˚C.

As the original technique was unable to detect carboxymefloquine, selected reaction monitoring (SRM) was used to identify carboxymefloquine by simultaneously searching for its m/z value, 308.01, as well as the m/z value for its daughter ion fragment 264.10. The LC-MS equipment consisted of a Thermo Scientific Vanquish UHPLC system (Thermo Fisher Scientific, San Jose, CA, USA) connected to a TSQ Altis Triple Quadrupole mass spectrometer (Thermo Fisher Scientific, San Jose, CA, USA). As with the earlier pseudo-targeted approach, the same column was used for chromatographic separation and the same software for instrument control and data processing. The eluents were water (+ 0.1% formic acid) (A) and methanol (+0.1% formic acid) (B). The run time was 10 minutes. For this method a gradient was used: 0 to 3 min B: 30–50%; 3 to 5 min B: isocratic 95%; and for 5 to 10 min B: 50–30%. The eluent flow rate was 0.30 mL/min. The injection volume was 1.0 μL and the column temperature was 45˚C. Heated-electrospray ionisation (H-ESI) parameters were: capillary voltage 3.5 kV positive mode and 2.5 kV negative mode, sheath gas 50 arbitrary units and auxiliary gas 10 arbitrary units. Carboxymefloquine was analysed in SRM mode with 25 V (positive) and 10 V (negative) collision energy. The SRM transition for carboxymefloquine was $308 \rightarrow 264$.

### In vitro $Cl_{int}$ calculations

In vitro $Cl_{int}$ was estimated by the substrate depletion method originally described by Obach (1999) [18]. Using this in vitro half-life ($t_{1/2}$) approach, the peak ratio of mefloquine / IS at t = 0 was set as 100% of mefloquine. Likewise, the peak ratios of the other time points were also converted to a percentage of the remaining mefloquine. These were then plotted as the natural log of remaining drug vs. incubation time whereby the slope of the regression line, represented as a rate constant (-k), was used for estimation of the in vitro $t_{1/2}$ = -0.693 / k. From this, in vitro $Cl_{int}$ = (0.693 / in vitro $t_{1/2}$) × (μL incubation volume / mg protein).

In determining in vitro $Cl_{int}$ through the substrate depletion method, if substrate depletion was < 15% of the initial value (at t = 0 minutes), depletion was deemed too slow to measure [20]. Consequently, under those circumstances in vitro $Cl_{int}$ could not be estimated due to accuracy and precision concerns [20, 27].

## Results

### Phase 1 reaction

Mefloquine phase I depletion was observed with respect to the feline and brush-tailed possum microsomes but not with the canine microsomes. Comparative chromatograms of mefloquine depletion at 30 mins by all microsomes and the negative control are illustrated in **Fig 1.**

Mefloquine depleted ≥ 30% over 60 mins when incubated with feline and possum microsomes. However there was no significant difference between the mean depletion (as a percentage) between these two species at any time-point (0, 30 and 60 min). In comparison, mefloquine depletion did not occur over 90 minutes when incubated with canine microsomes (**Fig 2**) with the actual depletion data provided as *S1 Table*. Tramadol, as the positive control, depleted > 70% and > 75% over the 60 minute incubation with both feline and canine microsomes, respectively.

The apparent in vitro $Cl_{int}$ (mean ± S.D.) of mefloquine by pooled feline and possum microsomes were 4.5 ± 0.35 and 18.25 ± 3.18 μL/min/mg protein, respectively whereas the in vitro $Cl_{int}$ of mefloquine by pooled canine microsomes could not be determined because substrate depletion was < 15% of the initial value (at t = 0 minutes).

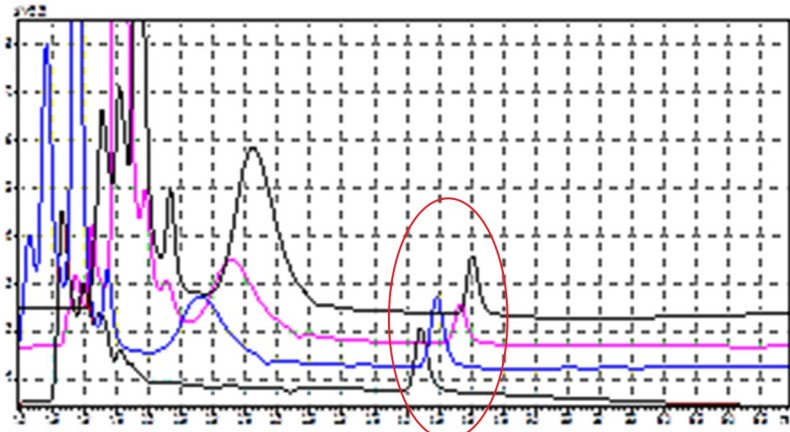

**Fig 1. Chromatograms showing mefloquine peaks (located inside red circle at a retention time of 4.3 mins) after 30 mins.** Brown trace = negative control i.e. incubation with no NADP cofactors, blue trace = canine microsome incubation, pink trace = possum microsome incubation and black trace = feline microsome incubation.

Based on the conditions used for HPLC analysis, a potential phase I metabolite appeared in the feline microsomal incubation samples between 2.4 and 2.55 minutes but was not observed in the canine samples. Although initially, LC-MS pseudo-targeting could not identify this peak, SRM later identified carboxymefloquine not only in feline samples but also in canine samples and negative controls.

## Phase II reaction

In the feline, canine and common brush-tailed microsomes, the mefloquine depletion was < 15% of the initial value (at time = 0 minutes) and thus too slow to determine (Table 1). The rate of ODMT depletion over 180 mins with canine and possum microsomes was > 15% while the ODMT depletion with feline microsomes was not detected by HPLC. However, a small ODMT glucuronide signal was detected in the feline samples by LC-MS [22].

## Discussion

This is the first reported study on the in vitro microsomal metabolism of mefloquine by feline microsomes. This screen provides an important first step in the description of mefloquine's pharmacokinetic (PK) and safety profile as a precursor to its use as an antiviral drug for treatment of FIP and FCV infections in cats.

Mefloquine was shown to undergo depletion with feline and possum microsomes and phase I metabolism cofactors. This is consistent with other observations that mefloquine undergoes in vitro phase I metabolism in other species, including humans [11, 12], rats [11, 28] and monkeys [11]. However, mefloquine has also been demonstrated to undergo very slow phase I metabolism with canine microsomes [11]. Fontaine *et al.* (2000) [11] reported that 85% of mefloquine remained in canine samples after a five-hour microsomal incubation whereas close to 100% of mefloquine was metabolized via phase I biotransformation by canine primary hepatocytes after 39-hours incubation. Whilst longer incubations are sometimes required (> 2 hours) [21], prolonged microsomal incubations were not undertaken in this

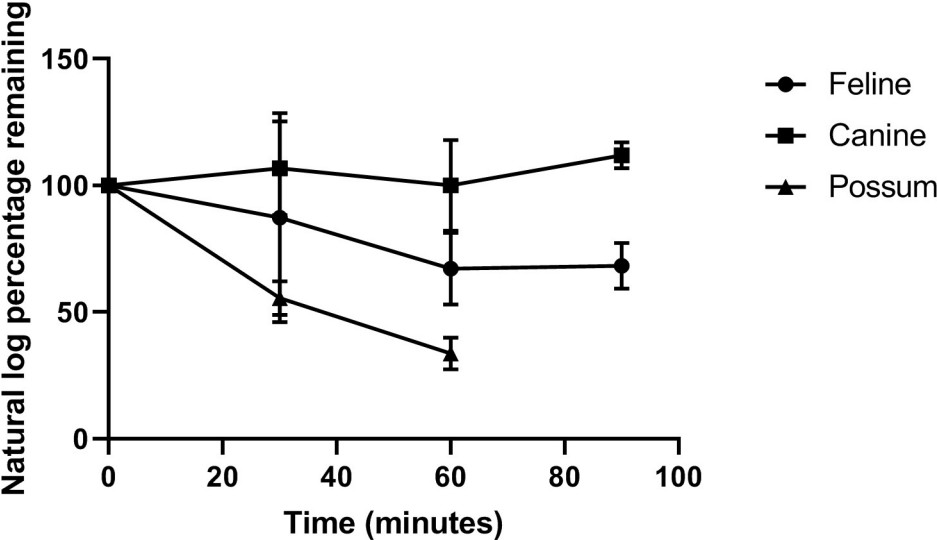

**Fig 2. Mefloquine depletion concentration (expressed as log substrate remaining) as a percentage vs. incubation time between species.** Microsome concentrations of 1.0 mg/mL for all species were incubated with 10 μM final concentrations of mefloquine. Duplicate samples were run for each time point for each species. The error bars represent the standard error of the mean (SEM).

**Table 1. Mefloquine and ODMT (as positive control) depletion as a percentage over time (mean ± S.D.) for the microsomal incubations of each species.**

| Time (minutes) | Feline microsomes | | Canine microsomes | | Possum microsomes | |
|---|---|---|---|---|---|---|
| | Mefloquine depletion (%) | ODMT depletion (%) | Mefloquine depletion (%) | ODMT depletion (%) | Mefloquine depletion (%) | ODMT depletion (%) |
| 0 | 100 ± 0.0 | 100 ± 0.0 | 100 ± 0.0 | 100 ± 0.0 | 100 ± 0.0 | 100 ± 0.0 |
| 60 | 108.95 ± 27.13 | 106.84[a] | 100 ± 0.0 | 84.44 ± 0.87 | 107.27 ± 0.13 | 36.19 ± 6.35 |
| 180 | 109.48 ± 1.69 | 118.96 ± 0.48 | 103.60 ± 9.95 | 63.51 ± 5.16 | 119.50 ± 0.44 | 11.63 ± 6.33 |
| 360 | 125.51 ± 4.46 | 118.59 ± 0.14 | 110.91 ± 9.66 | 49.99 ± 1.2 | 112.95 ± 6.95 | 2.78 ± 1.34 |

[a]Only one value at t = 60 minutes for the ODMT feline group was available.

study due to concerns of extended incubation times resulting in declining enzyme activity [20] and ultimately an underestimation of intrinsic clearance. Here, unchanged canine microsomal mefloquine depletion percentages greater than 100% indicated that no substrate depletion had occurred in these samples after 90 minutes. Although arguably the duration used in this study was short, canine microsomal incubations undertaken for more than 360 minutes likewise did not demonstrate mefloquine depletion. It is possible that some mefloquine depletion may have occurred but went undetected by HPLC due to assay insensitivity.

The use of tramadol as the phase I positive control confirmed that in vitro phase I hepatic metabolism occurred with both feline and canine microsomes. The advantage of using tramadol as the phase I positive control is that both mefloquine and tramadol are considered to undergo phase I metabolism catalysed by the same cytochrome P450 enzyme (CYP), e.g., the CYP3A subfamily in humans [11, 23]. However, in humans, tramadol is catalysed by CYP3A4 and CYP286 to NDMT. A schematic representation of tramadol metabolism in the human liver cell is found at https://www.pharmgkb.org/pathway/PA165946349 [23].

Mefloquine is reported to undergo in vitro phase I biotransformation to two major metabolites, carboxymefloquine (primary metabolite) and hydroxymefloquine (to a much lesser degree), in human, rat, monkey and dog models [11, 12]. Mefloquine phase I feline microsomal incubations generated an unknown peak that increased in amplitude over time. This peak was subsequently identified by SRM as carboxymefloquine but it was also identified by SRM in both the canine incubation samples and the negative controls. Carboxymefloquine is not considered a metabolite generated by the feline microsomes because of its presence in negative controls and phase I metabolism reactions are not associated with carboxylation. It is likely that identification of this substrate is associated with mefloquine instability from processing. This mefloquine instability could explain the slight reversal of mefloquine depletion with feline microsomes between 60 and 90 mins of incubation (*Fig 2*). Identification of hydroxymefloquine was not pursued as it is only produced in relatively minor amounts [11, 12] and its m/z value has not been established.

This study also demonstrated that mefloquine does not appear to undergo phase II conjugative glucuronidative metabolism when incubated by microsomes of cats, dogs and the common brush-tailed possum. Canine and common brush-tailed possum microsome incubations were used as the positive control for this phase II reaction as they have been shown to deplete ODMT to ODMT glucuronide [22]. This study supports the hypothesis that phase II conjugative metabolism may not be a route of mefloquine metabolism in any species. Fontaine *et al*. (2000) [11, 12] dismissed the contribution of phase II conjugative glucuronidation in the biotransformation of mefloquine because none of the unidentified mefloquine metabolites produced by the dog, rat and monkey hepatocyte studies were hydrolysed by *β*-glucuronidase. It is therefore likely that mefloquine does not undergo phase II conjugative glucuronidation and

will not present the issues associated with the administration of other drugs that undergo this route of metabolism to cats (e.g., acetaminophen/paracetamol) [13, 29, 30].

In vitro metabolism models, such as microsomal incubation, are an invaluable screening tool used prior to undertaking in vivo studies for a species in which the in vivo pharmacokinetic profile of a drug is unknown. For example, the microsomal incubation model provides information on a parent drug's rate of metabolism as well as some information on its metabolites. However, as microsomes only contain phase I drug metabolising enzymes (DMEs) such as CYPs, flavin-containing monooxygenases and phase II UGTs, they therefore do not represent the entire spectrum of DMEs [31]. Consequently, further experiments are required to elucidate other mefloquine elimination pathways.

The microsomal assay used here did have some limitations. For example, pooled microsomes were used for each species whereby three individuals were represented in a pool. Although this is not uncommon practice, use of pooled microsomes may have overlooked differences in one or more individual's metabolic capacity. Moreover, during the phase II metabolism studies, phosphate buffer (pH 7.4) was used as described by Slovak et al. (2017) [14]. However, the use of Tris buffer (as opposed to a phosphate buffer) has been shown to lead to an increase in the activity of some UGTs (such as UGT1A4 and UGT1A9) when incubated with human liver microsomes [32]. Yet, as this was a comparative species microsomal study, the use of phosphate buffer should have little consequence unless a species had significantly different concentrations and/or activities of hepatic UGTs. Furthermore, in contrast to the common brush-tailed possums microsomes that were harvested by the authors, the feline and canine microsomes were purchased from Europe since they could not be sourced locally. Although quality control information concerning activity for various substrates was provided by the manufacturer, the results may have been different if the microsomes from all three species underwent the same extraction, transportation and storage conditions. Finally, as the canine and feline microsomes only contained contributions from male subjects, the activity of CYP3A is recognized to have different activity for a substrate which is sex dependent [33], therefore further research evaluating sex differences in substrate intrinsic clearance may be required.

Mefloquine is recognised to result in a variety of side effects in multiple human body systems including psychiatric [34, 35], gastrointestinal [36, 37], cardiovascular [38], haematological [39] and dermatological manifestations [40]. If mefloquine is to be used as an antiviral for cats in the future, this study indicates that cautious administration is warranted if the feline patient has any degree of liver dysfunction given that the liver is the primary site of phase I metabolism.

Because mefloquine is an antimalarial, the in vivo human PK profile is known. Mefloquine and its metabolites are excreted slowly from the body through faeces and urine [38, 41] with a terminal elimination half-life of roughly three weeks in healthy subjects, 10 to 14 days in malaria patients with uncomplicated falciparum malaria [42, 43] and approximately 20 days in cases involving more severe malaria [43, 44]. Its only reported use in veterinary medicine has been as an antimalarial drug for raptors [45]. However, mefloquine's exact mechanisms of action as an antimalarial agent and/or as an antiviral agent are unknown [8, 46, 47].

Recently, antivirals have been developed that have been shown to be safe and efficacious against FIPV in naturally and experimentally infected cats. One of these agents is a broad spectrum coronavirus protease inhibitor, GC376 [48], while the other is an adenosine nucleoside analogue, GS-441524 [4, 5]. However, neither of these antivirals have obtained registration for veterinary use. Consequently, it remains important that investigations into a readily available drug such as mefloquine with anti-FIPV activity, is urgently required for this invariably fatal disease. This in vitro study provides information on mefloquine's hepatic metabolism in cats

and indicates that it does not result in delayed metabolism in cats. Further studies should be undertaken to describe mefloquine's PK profile in the cat as well as its efficacy in treating FIPV and FCV infected cats.

## Supporting information

**S1 Table.**
(XLSX)

## Acknowledgments

Professor Michael Court (Washington State University, College of Veterinary Medicine) provided invaluable insight into the design of the phase II glucuronidative metabolism assay. Likewise, Dr. Ben Crossett and Mr. David Maltby of the Mass Spectrometry Hub of the Charles Perkins Centre of The University of Sydney supplied technical assistance with the LC-MS investigation.

## Author Contributions

**Conceptualization:** Aaron Michael Izes, Benjamin Kimble, Merran Govendir.

**Data curation:** Benjamin Kimble.

**Formal analysis:** Aaron Michael Izes, Benjamin Kimble, Merran Govendir.

**Funding acquisition:** Jacqueline Marie Norris, Merran Govendir.

**Investigation:** Aaron Michael Izes, Benjamin Kimble.

**Methodology:** Benjamin Kimble, Merran Govendir.

**Project administration:** Benjamin Kimble, Jacqueline Marie Norris, Merran Govendir.

**Resources:** Jacqueline Marie Norris, Merran Govendir.

**Supervision:** Jacqueline Marie Norris, Merran Govendir.

**Writing – original draft:** Aaron Michael Izes, Merran Govendir.

**Writing – review & editing:** Benjamin Kimble, Jacqueline Marie Norris, Merran Govendir.

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
