## [Decision Letter · Decision Letter 0]

12 Feb 2020

PONE-D-20-00567

In vitro hepatic metabolism of mefloquine using microsomes from cats, dogs and the common brush-tailed possum (Trichosurus vulpecula)

PLOS ONE

Dear Dr. Merran Govendir

Thank you for submitting your manuscript to PLOS ONE. After careful consideration, we feel that it has merit but does not fully meet PLOS ONE’s publication criteria as it currently stands. Therefore, we invite you to submit a revised version of the manuscript that addresses the points raised during the review process.

ACADEMIC EDITOR: 

Please, authors must be clear about the microsome characterization and bring more description about the data found and increase this info.

We would appreciate receiving your revised manuscript by Mar 28 2020 11:59PM. To enhance the reproducibility of your results, we recommend that if applicable you deposit your laboratory protocols in protocols.io, where a protocol can be assigned its own identifier (DOI) such that it can be cited independently in the future. For instructions see: http://journals.plos.org/plosone/s/submission-guidelines#loc-laboratory-protocols

We look forward to receiving your revised manuscript.

Kind regards,

Carlos E. Ambrósio, Ph.D

Academic Editor

PLOS ONE

Journal Requirements:

1. Please upload a new copy of Figure 2 as the detail is not clear. Please follow the link for more information: http://blogs.PLOS.org/everyone/2011/05/10/how-to-check-your-manuscript-image-quality-in-editorial-manager/

Reviewers' comments:

Reviewer's Responses to Questions

**Comments to the Author**

1. Is the manuscript technically sound, and do the data support the conclusions?

Reviewer #1: Yes

2. Has the statistical analysis been performed appropriately and rigorously? 

Reviewer #1: No

3. Have the authors made all data underlying the findings in their manuscript fully available?

Reviewer #1: Yes

4. Is the manuscript presented in an intelligible fashion and written in standard English?

Reviewer #1: Yes

5. Review Comments to the Author

Reviewer #1: The investigation is interesting, well conducted (good controls) and regards a very important step for a possible treatment for cats with FIP infection, in the future. Major revision is indicated, because:

#line 74: Materials and Methods are detailed, but we can not find any statistical analysis. Is there any reason for this?

#line 289: No statistical analysis, comparing times and also groups (species).

6. PLOS authors have the option to publish the peer review history of their article (what does this mean?). If published, this will include your full peer review and any attached files.

Reviewer #1: No

---

## [Author Response · Author response to Decision Letter 0]

20 Feb 2020

Authors’ response to review of 

Re: PONE-D-20-00567

 In vitro hepatic metabolism of mefloquine using microsomes from cats, dogs and the common brush-tailed possum (Trichosurus vulpecula) 

The Authors would like to thank the Academic Editor and the Reviewer for their comments concerning this manuscript. 

The Academic Editor’s comments

• Please, authors must be clear about the microsome characterization and bring more description about the data found and increase this info. 

Accordingly, the Authors have added the following: 

• the sex of the brush-tail possums used (2 female and 1 male) see lines 86 to 87

• the activity of the canine and feline microsomes as determined by the manufacturer has been added to the Supplementary File (S1) 

We hope this information is acceptable to the Academic Editor, however if you want any additional information please provide us more details on what is required and we will try to provide what you are seeking. 

• As requested a new revised copy of Figure 2 has been uploaded 

The Reviewer’s comments 

• #line 74: Materials and Methods are detailed, but we can not find any statistical analysis. Is there any reason for this?

#line 289: No statistical analysis, comparing times and also groups (species).

The substrate depletion model used in this manuscript is unreliable when the rate of depletion over the incubation period is < 15%. Di, and Obach (2015). "Addressing the challenges of low clearance in drug research." AAPS J. 17: 352-357..

This is the case with the phase I depletion using canine microsomes. 

This is also the case with the phase II depletion with the three groups of microsomes. 

The only statistical analysis that could be undertaken was to compare the Phase I depletion of the feline microsomes verses the possum microsomes. The mefloquine depletion by species, and vs time point of feline vs possum microsomes (at 0 vs 30 and 60 mins) using a 2 way ANOVA was not significant, when the level of significance was accepted at P < 0.05 

Feline vs possum microsomes Species difference over 60 mins p = 0.1838; Time point differences (0, 30 and 60 mins) p = 0.1255

Therefore the following has been added to the manuscript:

Materials and methods

Lines 199-204 Statistical analyses 

Statistical data analysis was performed using GraphPad Prism 8.1.1 software (Graph Pad Software, Inc., CA, USA). The substrate depletion model is not reliable when the depletion is < 15% during the incubation period [20]. However if the depletion was greater than 15% then a two way ANOVA was performed to compare the mean depletion (%) between species at each time-point. If the ANOVA was significant Tukey’s multiple comparison tests were undertaken. Statistical significance was accepted at P < 0.05. 

Results see lines 268 – 269 

Mefloquine depleted > 30% over 60 mins when incubated with feline and possum microsomes. However there was no significant difference between the mean depletion (as a percentage) between these two species at any time-point (0, 30 and 60 min). In comparison, mefloquine depletion did not occur over 90 minutes when incubated with canine microsomes (Fig 2) with the actual depletion data provided as S1 Table. Tramadol, as the positive control, depleted > 70% and > 75% over the 60 minute incubation with both feline and canine microsomes, respectively.

---

## [Decision Letter · Decision Letter 1]

13 Mar 2020

In vitro hepatic metabolism of mefloquine using microsomes from cats, dogs and the common brush-tailed possum (Trichosurus vulpecula)

PONE-D-20-00567R1

Dear Dr. Govendir,

We are pleased to inform you that your manuscript has been judged scientifically suitable for publication and will be formally accepted for publication once it complies with all outstanding technical requirements.

With kind regards,

Carlos E. Ambrósio, Ph.D

Academic Editor

PLOS ONE

Additional Editor Comments (optional):

Reviewers' comments:

Reviewer's Responses to Questions

**Comments to the Author**

1. If the authors have adequately addressed your comments raised in a previous round of review and you feel that this manuscript is now acceptable for publication, you may indicate that here to bypass the “Comments to the Author” section, enter your conflict of interest statement in the “Confidential to Editor” section, and submit your "Accept" recommendation.

Reviewer #1: All comments have been addressed

2. Is the manuscript technically sound, and do the data support the conclusions?

Reviewer #1: Yes

3. Has the statistical analysis been performed appropriately and rigorously? 

Reviewer #1: Yes

4. Have the authors made all data underlying the findings in their manuscript fully available?

Reviewer #1: Yes

5. Is the manuscript presented in an intelligible fashion and written in standard English?

Reviewer #1: Yes

6. Review Comments to the Author

Reviewer #1: The authors have made all the suggestions requested and the manuscript is now acceptable for publication.

7. PLOS authors have the option to publish the peer review history of their article (what does this mean?). If published, this will include your full peer review and any attached files.

Reviewer #1: No